# Design and Characteristic Analysis of Cross-Capacitance Fuel-Level Sensor

**DOI:** 10.3390/s18113984

**Published:** 2018-11-16

**Authors:** Jing Yu, Hang Yu, Dongsheng Li

**Affiliations:** School of Metrology and Measurement Engineering, Jiliang University, Hangzhou 310018, China; hebyuhang@163.com (H.Y.); lidongsheng@cjlu.edu.cn (D.L.)

**Keywords:** principle cross capacitance, fuel level measurement, level sensor, single-tube sensor

## Abstract

A cross-capacitance liquid level sensor is based on the principle of cross capacitance. This study designed a new single-tube cross-capacitance fuel-level sensor. The fuel-level measurement model is established for a single-tube cross-capacitive sensor, and the relationship between the measured liquid level and sensor output capacitance is derived. The characteristics of the sensor were tested experimentally. The experimental results demonstrate that the linearity error of the liquid-level sensor of the single-tube calculation for the spacecraft is ±0.48%, the repeatability error is ±0.47%, and the hysteresis error is ±0.68%. The cross-capacitive fuel-level sensor developed in this study can be used in the fuel tank of spacecrafts owing to its low weight and high precision.

## 1. Introduction

As a reference for impedance, cross-capacitance is the highest measurement level in the field of electromagnetic metrology, except for quantum voltage and quantum resistance. The standard measurement uncertainty can reach to 10^−8^ [1,2]. The output value of cross-capacitance is only related to axial length. In principle, only one-dimensional length measurement is required; thus, it is easy to obtain higher accuracy [3]. Cross-capacitance can also be used in the development of new sensors owing to less error sources and high stability. Rae Duk Lee used cross-capacitance to measure the dielectric constant of a liquid with a standard uncertainty of ±0.02% [4]. New sensors based on cross-capacitance principle can replace conventional capacitive sensors. Processing and assembly errors can be reduced. Measurement results are independent of the electrode diameter, and measurement accuracy can be improved [5]. A single-tube structure prevents the problem of deep hole machining and the coaxial assembly of inner and outer electrodes. The size and weight of sensors are reduced by coaxial electrodes. High precision and a simple structure can improve quality, reduce the amount of materials used, and reduce costs. The new sensor can be used in aviation, aerospace, marine, automotive and other fuel level measurement fields [6]. Particularly in spacecraft propulsion technology, modern launch vehicles have started using nontoxic and nonpolluting liquid fuels with high propulsion ratio, adjustable thrust, and multiple ignition points [7,8]. Real-time and accurate measurement of the liquid level of the spacecraft fuel tank can ensure the optimal utilization of propellants, while ensuring the normal and safe operation of spacecraft engines. This has considerable practical significance for improving the effective carrying capacity and efficiency of rockets [9]. The capacitance per unit length is independent of the diameter, and the total capacitance is generally related to axial length. Cross-capacitance is used to develop new sensors owing to its advantages of fewer error sources and high stability [10,11].

In this study, we design a single-tube liquid-level sensor based on the principle of cross capacitance. The sensor is suitable for the real-time detection of the liquid level [12]. A fuel-level measurement model is established for the developed sensor, and the relationship between the measured liquid level and sensor output capacitance is derived. The characteristics of the sensor are experimentally tested, and the linearity, repeatability, and hysteresis of the designed sensor are examined.

## 2. Principle of Cross-Capacitance Liquid-Level Sensor

The principle of the cross-capacitive liquid-level sensor is based on the principle of cross-capacitance where the total capacitance per unit length of a cross-capacitor has high stability [13]. The mechanical structure of the sensor is shown in Figure 1.

He cross-capacitive liquid-level sensor in a spacecraft fuel tank comprises a single tube composed of an insulating body and a metal layer. This is different from the conventional capacitive liquid-level sensor [14]. The insulation body is a cylindrical hollow structure for holding liquid fuel, and it is made of quartz. The outer wall of the quartz tube is coated with a metal layer-copper film. Four tiny gaps are etched at intervals of 90° on the circumference of the copper film, and each small gap corresponds to a central equal angle, *α*, *β*, *γ*, *δ*, *α* = *β* = *γ* = *δ* << 90°. In this design, the angle is set as 2.6°. Four copper films sandwiched between two small gaps form four plates. Each plate corresponds to a central angle of *a_n_* ≈ 90° (*n* = 1, 2, 3, 4). The two plates corresponding to central angles *a*_1_ and *a*_3_ form a pair of capacitive outputs, and the two plates corresponding to central angles *a*_2_ and *a*_4_ form another pair of capacitive outputs, thereby forming a crossover capacitor according to the principle of cross capacitance.

Let us assume that the axial length of the plate is *l*, then [15]
1)The output capacitance of the empty tube is given by
(1)$$C_{a}=\frac{\varepsilon_{0}\varepsilon_{a}}{\pi }\ln2\cdot l\cdot (1+k_{a})$$2)The output capacitance of the full tube is given by
(2)$$C_{l}=\frac{\varepsilon_{0}\varepsilon_{l}}{\pi }\ln2\cdot l\cdot (1+k_{l})$$3)When the level of liquid fuel is *l_x_*, the cross-capacitive level sensor can be assumed as two upper and lower cross-capacitors. The upper part comprises cross-capacitor 1 with height *l* − *l_x_*, and the lower part comprises cross-capacitor 2 with liquid level *l_x_*. The medium to be measured is the liquid fuel to be tested. The capacitances of the two cross-capacitors are given by
(3)$$C_{a}^{\prime }=\frac{\varepsilon_{0}\varepsilon_{a}}{\pi }\ln2\cdot (l-l_{x})\cdot (1+k_{a})$$
(4)$$C_{l}^{\prime }=\frac{\varepsilon_{0}\varepsilon_{l}}{\pi }\ln2\cdot l_{x}\cdot (1+k_{l})$$
where $\varepsilon_{a}$ and $\varepsilon_{l}$ are the dielectric constants of air and the liquid fuel to be tested respectively. $k_{a}$ is the correction coefficients of air. $k_{l}$ is the correction factor of the output capacitor in a medium with dielectric constant $\varepsilon_{l}$. When an insulating tube is placed in a single-tube cross capacitor, there is a certain difference between the output capacitors, and the correction coefficients can be used.

When the measured liquid level is *l_x_*, the total output capacitance is given as follows:(5)$$C_{x}=C_{a}^{\prime }+C_{l}^{\prime }$$

Thus, the measured liquid level is given by
(6)$$l_{x}=\frac{\left(C_{x}-C_{a}\right)}{C_{l}-C_{a}}\cdot l$$

Based on the capacitance principle, the fuel-level measurement model only needs to measure the capacitor outputs of the empty tube, full tube, and current liquid level. The current liquid level of the measured liquid can be calculated in combination with the accurate numerical values along the axial length of the plates. The error in the measurement model depends only on the axial length of the plate and the dielectric constant of the liquid to be measured. The accuracy of the liquid level measurement can be improved, and the single-tube lightweight design provides the possibility of reducing the weight.

## 3. Design of a Capacitive Fuel Sensor

According to the mathematical model of the single-tube calculation of the capacitor structure, the single-tube capacitive fuel-level sensor is applied to solve the structural defects of the conventional capacitive liquid-level sensor. The single-tube structure of the capacitive fuel-level sensor is shown in Figure 2. The outer diameter, inner diameter, wall thickness, and height are 25 mm, 21 mm, 2 mm, and 300 mm, respectively. 

The designed sensor uses a quartz tube as a support structure, and a metal plating layer is plated on the outer wall of the quartz tube as an electrode. The metal plating layer is divided into a protective electrode by a scribe line and two sets of high electrodes and low electrodes. The high and low electrodes constitute a calculated capacitance structure to measure the fuel level. The protective electrodes can reduce the edge effect and improve sensitivity. 

Because the coating thickness is low, the outer diameter *D*_2_ of the capacitor and the inner diameter *D*_1_ of the insulating tube can be regarded as equal. The cross-sectional area of the sensor can be simplified as shown in Figure 3. 

When the capacitor is placed in air (when the effective length is *l*), if *R*_2_ = *R*_3_, the output capacitance under different conditions can be reduced to
(7)$$\begin{array}{ll} C_{a}^{\prime } & =C_{a}(1+k_{a})\\ & =\frac{\varepsilon_{0}\varepsilon_{a}}{\pi }\ln2\cdot \left\{1+\frac{{\left(\frac{\varepsilon_{w}}{\varepsilon_{a}}\right)}^{2}-1}{2\left(\frac{\varepsilon_{w}}{\varepsilon_{a}}\right)\ln2}\ln\frac{2\left[1+{\left(\frac{R_{1}}{R_{3}}\right)}^{4}\right]}{{\left[1+{\left(\frac{R_{1}}{R_{3}}\right)}^{2}\right]}^{2}}\right\}\cdot l \end{array}$$

When the capacitor is placed in a medium with dielectric constant $\varepsilon_{l}$ (when the effective length is *l*)
(8)$$\begin{array}{ll} C_{l}^{\prime } & =C_{l}(1+k_{l})\\ & =\frac{\varepsilon_{0}\varepsilon_{l}}{\pi }\ln2\cdot \left\{1+\frac{{\left(\frac{\varepsilon_{w}}{\varepsilon_{l}}\right)}^{2}-1}{2\left(\frac{\varepsilon_{w}}{\varepsilon_{l}}\right)\ln2}\ln\frac{2\left[1+{\left(\frac{R_{1}}{R_{3}}\right)}^{4}\right]}{{\left[1+{\left(\frac{R_{1}}{R_{3}}\right)}^{2}\right]}^{2}}\right\}\cdot l \end{array}$$
where *R*_1_ and *R*_2_ are the inner and outer radii of the insulating tube, respectively, *R*_3_ is the radius of the outer wall of the sensor after coating, *k_a_* is the correction coefficient of the output capacitor in air, $\varepsilon_{w}$ is the dielectric constant of the insulating tube wall, and *k_l_* is the correction factor of the output capacitor in a medium with dielectric constant $\varepsilon_{l}$.

When the single-tube capacitive fuel level sensor is measured, a part of the sensor is immersed in the liquid fuel to be tested (dielectric constant $\varepsilon_{l}$) and a part of it is exposed to air. During the measurement process, when the sensor measurement range is *l* and the measured fuel level is *l_x_*, the total capacitance *C_x_* output by the sensor is the sum of the capacitance $C_{a1}^{\prime }$ of the portion of sensor exposed to air and the capacitance $C_{l1}^{\prime }$ of the portion of sensor immersed in the fuel to be tested.
(9)$$C_{a1}^{\prime }=\frac{\varepsilon_{0}\varepsilon_{a}}{\pi }\ln2\left(1+k_{a}\right)\left(l-l_{x}\right)$$
(10)$$C_{l1}^{\prime }=\frac{\varepsilon_{0}\varepsilon_{l}}{\pi }\ln2\left(1+k_{l}\right)l_{x}$$

The output capacitance is:(11)$$\begin{array}{ll} C_{x} & =C_{a1}^{\prime }+C_{l1}^{\prime }\\ & =\frac{\varepsilon_{0}\varepsilon_{a}\ln2}{\pi }\cdot (1+k_{a})\cdot (l-l_{x})+\frac{\varepsilon_{0}\varepsilon_{l}\ln2}{\pi }\cdot (1+k_{l})\cdot l_{x}\\ & =\frac{\varepsilon_{0}\varepsilon_{a}\ln2}{\pi }\cdot (1+k_{a})\cdot l-\frac{\varepsilon_{0}\varepsilon_{a}\ln2}{\pi }\cdot (1+k_{a})\cdot l_{x}+\frac{\varepsilon_{0}\varepsilon_{l}\ln2}{\pi }\cdot (1+k_{l})\cdot l_{x}\\ & =C_{a}^{\prime }-\frac{\varepsilon_{0}\varepsilon_{a}\ln2}{\pi }\cdot (1+k_{a})\cdot l_{x}+\frac{\varepsilon_{0}\varepsilon_{l}\ln2}{\pi }\cdot (1+k_{l})\cdot l_{x} \end{array}$$

Thus
(12)$$\begin{array}{ll} (C_{x}-C_{a}^{\prime })\cdot l & =\frac{\varepsilon_{0}\varepsilon_{l}\ln2}{\pi }\cdot (1+k_{l})\cdot l\cdot l_{x}-\frac{\varepsilon_{0}\varepsilon_{a}\ln2}{\pi }\cdot (1+k_{a})\cdot l\cdot l_{x}\\ & =\left(C_{l}^{\prime }-C_{a}^{\prime }\right)\cdot l_{x} \end{array}$$

Therefore, the measured fuel level is: (13)$$\begin{array}{ll} l_{x} & =\frac{\left(C_{x}-C_{a}^{\prime }\right)}{\left(C_{l}^{\prime }-C_{a}^{\prime }\right)}\cdot l\\ & =\frac{l}{C_{l}^{\prime }-C_{a}^{\prime }}C_{x}-\frac{C_{a}^{\prime }l}{C_{l}^{\prime }-C_{a}^{\prime }} \end{array}$$

It can be seen that the measured fuel level is only affected by the dielectric constant of the fuel to be tested and the inner and outer diameters and ranges of the sensor. The measured fuel level varies linearly with the sensor output capacitance.

## 4. Experimental Characteristics of the Sensor

In this study, absolute ethanol with a purity of 99.8% was used as the fuel to be tested. The sensor was fixed by a steel frame to keep it stable. Insulation measures were applied at the position where the sensor was in contact with the fixed frame to prevent unnecessary external interference.

The output capacitance of the developed sensor was measured by a capacitance measuring instrument (TH2617). The minimum resolution of TH2617 is 0.0001 pF, and its measurement accuracy is 0.05%. During the measurement process, the sensor was kept fixed and the liquid level of the measured fuel was varied. According to the accurate measurement of the liquid level, two sets of relative measurement electrode output capacitances, *C*_1_ and *C*_2_, were recorded, the average, *C_p_*, of these values was calculated. The change in liquid level could be monitored in real time. A good electrical conductor (stainless steel) was used as the outer casing of the developed sensor, which closed all sensing elements and performed reliable grounding simultaneously. Shielded cables insulated from each other were used as wires and grounded at the metal mesh. The measuring circuit part should also be shielded and grounded accordingly.

### 4.1. Linearity Experiment

When the measured fuel level is stable, the relationship between the sensor output capacitance and measured fuel level is linear. The input and output curves of the sensor were measured during the experiment and then fitted and linearized. Linearity error is expressed by relative error *γ*_L_ as
(14)$$\gamma_{L}=\pm \left(\Delta_{L\max}/y_{\mathrm{FS}}\right)\times 100\%$$
where $\Delta_{L\max}$ is the maximum nonlinear error and $y_{\mathrm{FS}}$ is the full scale output. 

The relationship between the liquid level and output capacitance is shown in Figure 4. The experimental results show that the capacitances, *C*_1_ and *C*_2_, of the two groups change linearly between 0.6 pF and 17 pF as the liquid level increases to 200 mm. As shown in Figure 5, a better linear characteristic curve can be obtained by considering the average of *C*_1_ and *C*_2_ as the output, and the linearity error can be calculated as ±0.48%.

### 4.2. Repeatability Experiment

Calculating the inconsistency in the characteristic curve obtained by the capacitive fuel-level sensor in the same measurement direction (forward or reverse) for multiple consecutive full-scale experiments can help in deriving the repeatability error for the developed sensor. This error is typically expressed as follows:(15)$$\gamma_{R}=\pm \left(\Delta_{R\max}/y_{\mathrm{FS}}\right)\times 100\%$$
where $\Delta_{R\max}$ is the largest repeatability error in forward and reverse strokes. The characteristic curve of the fuel level and the average output capacitances, *C*_1_ and *C*_2_, during the forward and reverse strokes are shown in Figure 6 and Figure 7, respectively.

The experimental results show that the repeatability error for the forward stroke is ±0.69% when the liquid level varies from 0 to 200 mm. After system compensation, the data with large repeatability error in the edge range are removed, and the sensor repeatability error is obtained as ±0.47%. The repeatability error for the return stroke is ±0.59% when the liquid level varies from 0 to 200 mm. After system compensation, the data with large repeatability error in the edge range are removed, and the sensor repeatability error is obtained as ±0.42%. Thus, the repeatability error for the sensor is ±0.47%.

### 4.3. Hysteresis Test

During the experiment, the misalignment between the input and output curves of the forward and return strokes is the hysteresis error for the capacitive fuel-level sensor. This error is represented as follows: (16)$$\gamma_{H}=\pm \left(1/2\right)\left(\Delta_{H\max}/y_{\mathrm{FS}}\right)\times 100\%$$
where $\Delta_{H\max}$ is the maximum difference between the positive and negative strokes. The relationship between the fuel level and three average output capacitances is shown in Figure 8. The experimental results show that the maximum hysteresis error for the sensor is ±0.68% when the liquid level varies from 0 to 200 mm.

## 5. Conclusions

1)Based on the cross-capacitance principle, we designed a new single-tube cross-capacitive fuel-level sensor.2)A fuel-level measurement model was established for the developed sensor, and the relationship between the measured liquid level and sensor output capacitance was derived. The feasibility of applying the capacitance principle to the measurement of fuel level was theoretically proved.3)The characteristics of the sensor were tested experimentally, and the linearity, repeatability, and hysteresis of the designed sensor were examined. The experimental results showed that the linearity error in the single-tube calculation for the liquid-level sensor was ±0.48%, the repeatability error was ±0.47%, and the hysteresis error was ±0.68%.4)The cross-capacitive fuel-level sensor developed in this study provides not only a new idea for the application of the capacitance principle to sensors but also a basis for the development of high-precision liquid-level sensors.

## Figures and Tables

**Figure 1 sensors-18-03984-f001:**
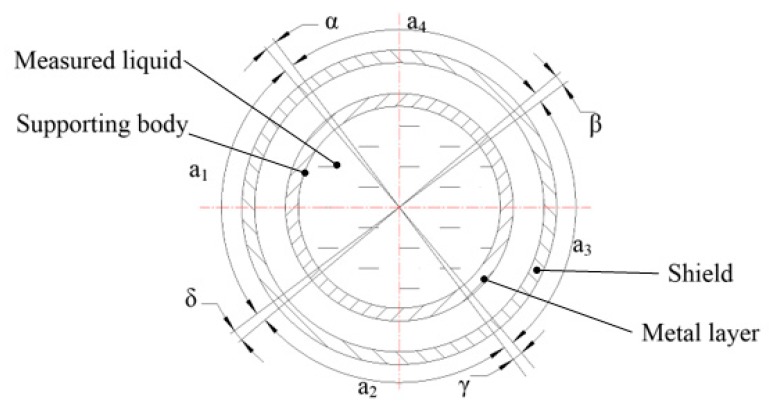
Principle of cross-capacitive liquid-level sensor.

**Figure 2 sensors-18-03984-f002:**
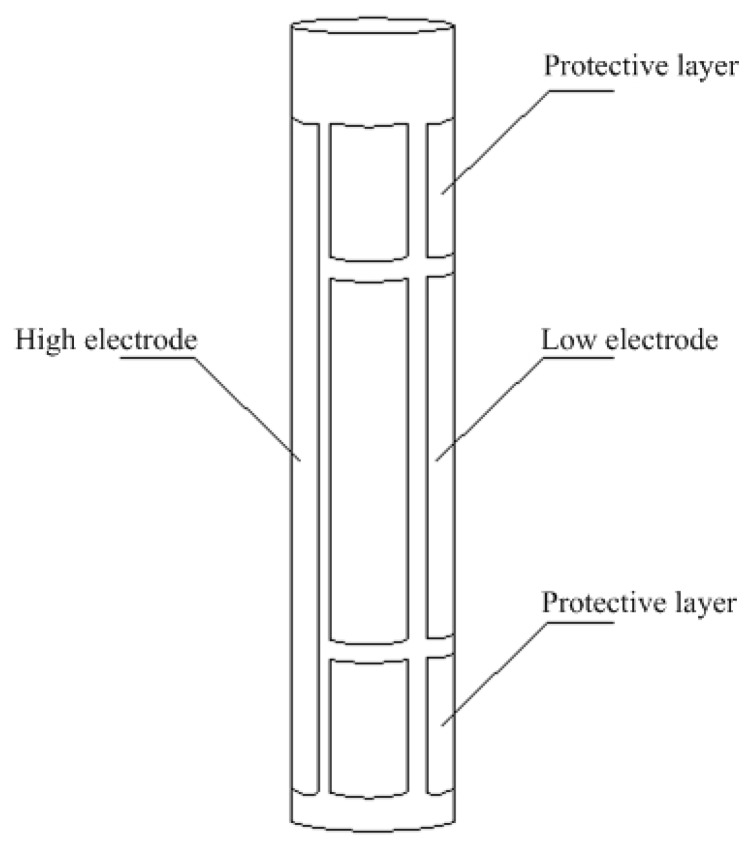
Schematic of the fuel-level sensor.

**Figure 3 sensors-18-03984-f003:**
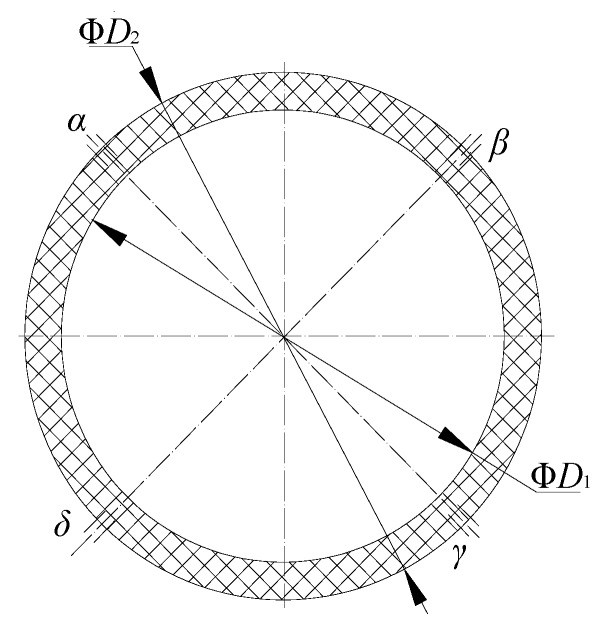
Simplified single-tube calculation capacitor.

**Figure 4 sensors-18-03984-f004:**
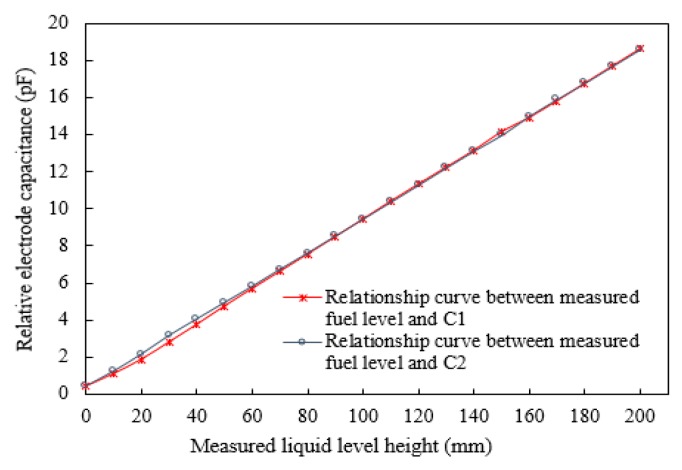
Relationship between liquid level and output capacitance.

**Figure 5 sensors-18-03984-f005:**
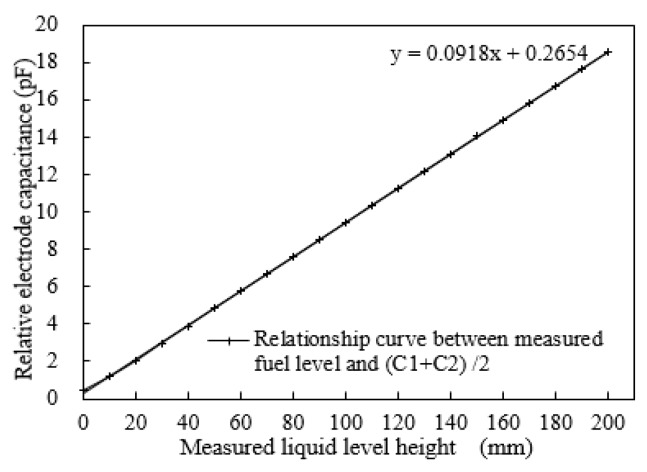
Characteristic curve of liquid level and output average capacitance.

**Figure 6 sensors-18-03984-f006:**
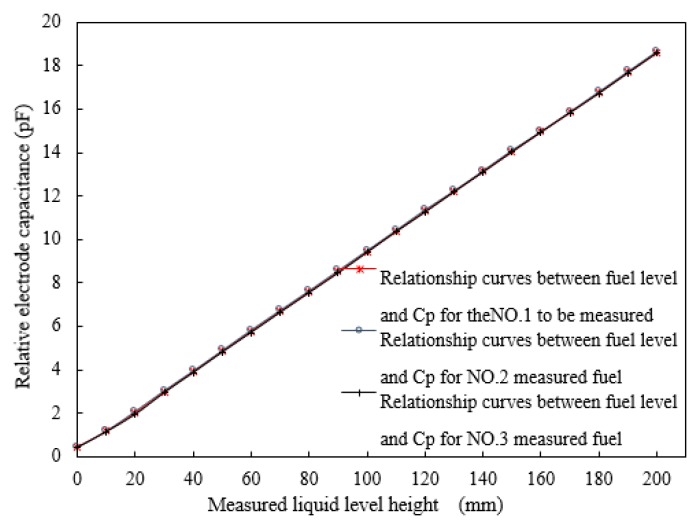
Relationship between fuel level and average output capacitance of the *C_P_* during the forward stroke.

**Figure 7 sensors-18-03984-f007:**
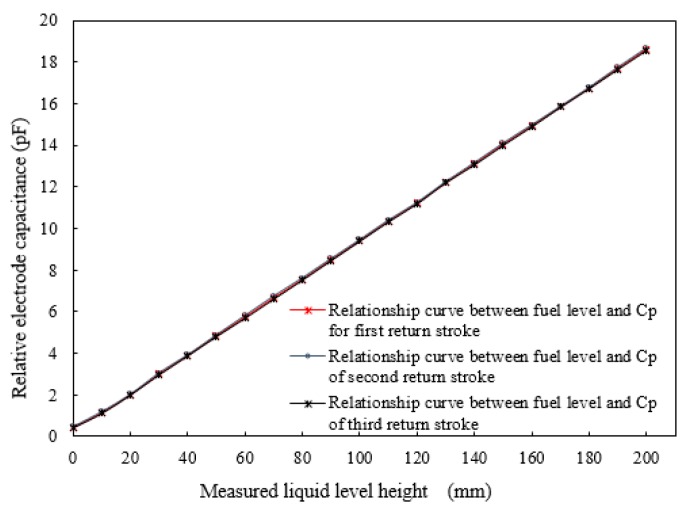
Relationship between fuel level and average output capacitance of the *C_P_* during the reverse stroke.

**Figure 8 sensors-18-03984-f008:**
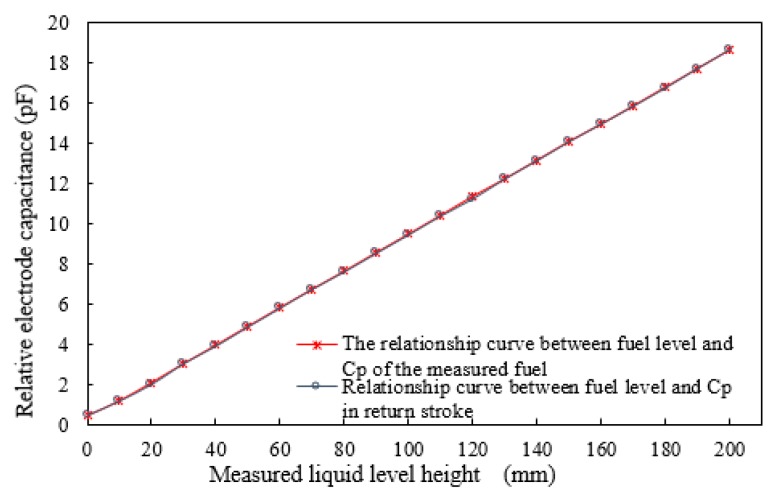
Characteristic curve of fuel level and average output capacitance *C_P_*.

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
