# Peer review of "Design and Characteristic Analysis of Cross-Capacitance Fuel-Level Sensor"

_sensors, 2018, doi:10.3390/s18113984_

Round 1
Reviewer 1 Report
Round1:
The
paper presents an interesting approach to measure the fuel tank level,
the result indicates that the designed cross-capacitance sensor has the
capability to provide an accurate measurement. However, BOTH Abstract
and Conclusions contain the results of ±0.2% in model error, ±0.047%
in repeatability error and ±0.44% in hysteresis error, NONE of these values can be found in the main content.
Also, adding model validation rather than only the calibration will
make the statement stronger. Besides, there are numerous small mistakes
that need to be fixed. I have marked most of them in the PDF file.
Round2:
The repeatability error for the sensor is ±0.47% in the 4.2, but the value in conclusion and abstract is ±0.047%
Lack of discussions

Author Response
Round 1:
1) BOTH Abstract and Conclusions contain the results
of ±0.2% in model error, ±0.047% in repeatability error and ±0.44% in
hysteresis error, NONE of these values can be found in the main content.
Also, adding model validation rather than only the calibration will
make the statement stronger.
Response: We recalculated the results
and observed the following results. The linearity error in the
single-tube calculation for the liquid-level sensor was ±0.48%, the
repeatability error was ±0.047%, and the hysteresis error was ±0.68%.
Both the abstract and the conclusions are modified.
2)Besides, there are numerous small mistakes that need to be fixed. I have marked most of them in the PDF file.
Response:
(1) Some unclarified sentences and references have been modified.
(2)
The true values of the angle are provided. The angle; α, β, γ, δ, α = β
= γ = δ << 90°. In this design, the angle is set as 2.6°.
(3)
The two opposing plates are shown in figure 1. The two plates
corresponding to central angles a1 and a3 form a pair of capacitive
outputs, and the two plates corresponding to central angles a2 and a4
form another pair of capacitive outputs, thereby forming a crossover
capacitor according to the principle of cross-capacitance.
(4) The capacitance-measuring instrument (TH2617) has been listed.
(5) Figure 5 has been modified, wherein the unit of the horizontal axis is provided.
(6) Some grammatical oversights have been modified.
Round 2:
Response: The repeatability error for the sensor is ±0.47%, we mistake the data in conclusion and abstract. It has been modified in the article.The repeatability error is calculated by system compensation.
Reviewer 2 Report
The authors present a quite comprehensive study of their proposed sensor design. There are no obvious flaws in the (clearly) presented calculations. I have only few comments and thoughts:
line 94: "walk thickness" should be "wall thickness"
The authors claim that their sensor design would be particularly suitable for application in spacecraft due to its low weight. However, the accuracy measurements were carried out with a TH2617, which has a weight of about 3.5 kg. The authors should at least mention that the sensor setup does not only consist of the small cross-capacitor itself, but also requires a not-so-light-weight capacitance readout system (although not necessarily a TH2617).
Another question about the application in spacecrafts: All calculations are based on a clear definition of the liquid level in the sensor tube. Did the authors take into account the behavior of the liquid at zero gravity? At first glance I would assume that the movement of the liquid inside the tube will somehow affect the measurement accuracy. The authors should discuss this, even if it may not play a role, as I would expext that other readers might have similar thoughts.
Author Response
line 94: "walk thickness" should be "wall thickness"
Response: It has been modified.
The authors claim that their sensor design would be particularly suitable for application in spacecraft due to its low weight. However, the accuracy measurements were carried out with a TH2617, which has a weight of about 3.5 kg. The authors should at least mention that the sensor setup does not only consist of the small cross-capacitor itself, but also requires a not-so-light-weight capacitance readout system (although not necessarily a TH2617).
Response: Th2617 is used the measure the static characteristics of the sensor. When the sensor is used in spacecraft, TH2617 is not necessary. The actual capacitance readout of the sensor is based on a capacitance conversion chip. This part of the capacitance conversion chip is in another article.
Another question about the application in spacecrafts: All calculations are based on a clear definition of the liquid level in the sensor tube. Did the authors take into account the behavior of the liquid at zero gravity? At first glance I would assume that the movement of the liquid inside the tube will somehow affect the measurement accuracy. The authors should discuss this, even if it may not play a role, as I would expext that other readers might have similar thoughts.
Response: Thank you for your advice, actually, the liquid level is measured in the conventional environment, we will consider about the liquid at zero gravity in our next experiment.